# An empirical QPE method based on polarimetric variable adjustments

Jungsoo Yoon<sup>1</sup>, Jong-Sook Park<sup>1</sup>, Hae-Lim Kim<sup>1</sup>, Mi-Kyung Suk<sup>1</sup>, Kyung-Yeub Nam<sup>1</sup> <sup>1</sup>Weather Radar Center, Korea Meteorological Adimistration *Correspondence to*: Jong-Sook Park (jspark9957@gmail.com)

- Abstract. This study presents an empirical method for optimizing polarimetric variables in order to improve the accuracy of dual-polarization radar rainfall estimation using data derived from radars operated by different agencies. The empirical method was developed using the Yong-In Testbed (YIT) radar operated by the Korea Meteorological Administration (KMA). The method is based on the determination of relations between polarimetric variables. Relations for  $Z Z_{DR}$  and  $Z K_{DP}$  are derived from the measurements of a two-dimensional video disdrometer installed about 30 km away from the YIT radar.
- These relations were used to adjust the polarimetric variables of the dual-polarization constant altitude plan position indicator (CAPPI) at a height of 1.5 km. The CAPPI data with the adjusted polarimetric variables were used to estimate rainfalls using three different radar rainfall estimation algorithms. The first algorithm is based on *Z*, the second on *Z* and  $Z_{DR}$ , and the third on *Z*,  $Z_{DR}$ , and  $K_{DP}$ . The accuracy of the radar-estimated rainfall was then assessed using raingauge observations. Three rainfall events with more than 40 mm of maximum hourly rainfall were shown to have the best estimation when the
- method using *Z*,  $Z_{DR}$ , and  $K_{DP}$  was used. However, stratiform precipitation events were better estimated by the algorithm using *Z* and  $Z_{DR}$ . The method was also applied to the data of three radars that belong to KMA and the Ministry of Land, Infrastructure, and Transport. The evaluation was done for six months (May–October) in 2015. The results show an improvement in radar rainfall estimation accuracy for stratiform, frontal, and convective precipitation from approximately 50 % to 70 %.

# 20 1 Introduction

() ()

Quantitative precipitation estimation (QPE) using radar data is sensitive to data quality and precipitation variability such as, rainfall types and their temporal and spatial changes. In South Korea, the agencies of three different ministries operate radars: the Korea Meteorological Administration (KMA) in the Ministry of Environment, the Han River Flood Control Office in the Ministry of Land, Infrastructure, and Transport (MOLIT), and the Korean Air Force Weather Group in the

- Ministry of National Defense. Each agency operates their own radars for different purposes such as observing weather, hydrology, and military operational weather. In addition, the radars are operated using different observation strategies, and the data undergoes different processing algorithms and other relevant techniques. Because of these differences, there is a wide range of accuracy in the observed radar data as well as the composite images, which consist of cross-governmental radar measurements. These factors have greatly impacted radar-based quantitative rainfall estimation based on cross-
- governmental radar measurements. Obtaining radar data with a fairly high level of accuracy is a common concern of the

three ministries. Thus, an agreement to harmonize weather and hydrological radar products was made by the three ministries in 2010, called "Development and application of Cross governmental dual-pol radar harmonization." The aim of the agreement is to produce uniformly good quality radar data and radar-based rainfall estimations, as high quality data is one of the most important advantages of dual-polarization radars. KMA has played a leading role in achieving these aims, which are the motivation for this study.

- It is believed that dual-polarization radar can better estimate rainfall, and many researchers have developed better radar rainfall estimation algorithms using polarimetric variables. Seliga and Bringi (1976) derived drop size distribution from  $Z_{DR}$ and also demonstrated that rainfall can be estimated by  $Z_{DR}$ . Others have proved that rainfall estimations using  $Z_{DR}$  are better than those using the conventional Z–R relation (Seliga et al., 1986; Aydin et al., 1987). In addition, Humphrise (1974) showed that specific differential phase ( $K_{DP}$ ) was linearly related to the rain rate, and rainfall estimation algorithms using  $K_{DP}$  were suggested by other researchers (Jameson, 1985; Sachidananda and Zrnić, 1986; Chandra et al, 1990). Later, Jameson (1991) suggested a rainfall estimation algorithm using both  $Z_{DR}$  and  $K_{DP}$ . Ryzhkov and Zrnić (1995a) found that algorithms using  $Z_{DR}$  and  $K_{DP}$  were better than other algorithms. The Colorado State University (CSU) ICE algorithm and Joint POLarization Experiment (JPOLE) algorithm are synthesized methods that selectively use Z,  $Z_{DR}$ , and  $K_{DP}$  with respect to the range of polarimetric variables or rain rate (Ryzhkov et al., 2003; Cifelli et al., 2011).
- Of the polarimetric variables, *Z*, which is a backscattered variable, can be affected by attenuation or partial beam blockage. Because of this feature, rainfall estimated using *Z* is generally lower than raingauge rainfall measurements (Austin, 1987; Ryzhkov and Zrnić, 1995b). Variable  $Z_{DR}$ , which is the ratio between horizontal and vertical reflectivity, is related to the axis ratio of hydrometeors (Zrnić and Ryzhkov, 1999; Straka et al., 2000; Frech and Steinert, 2015). When the radar antenna
- points vertically upward, because the shape of a hydrometeor is curved in the direction of the radar beam,  $Z_{DR}$  has to be 0 dB. Gorgucci et al. (1999) suggested a  $Z_{DR}$  calibration method using this concept. Variable  $K_{DP}$ , defined by the range derivative of differential phase shift  $\Phi_{DP}$ , is a propagation variable and not affected by attenuation or partial beam blockage (Ryzhkov and Zrnić, 1995a; Zrnić and Ryzhkov, 1999). Hence,  $K_{DP}$  can offer more accurate high rain rate estimations (Sachidananda and Zrnić, 1986; Chandarsekar et al., 1990).
- It is widely accepted that the three polarimetric variables Z,  $Z_{DR}$ , and  $K_{DP}$  are related to each other (Leitao and Watson, 1984; Aydin et al., 1986; Ryzhkov and Zrnić, 1996; Straka et al., 2000). Straka et al. (2000) classified hydrometeors with respect to the domains of polarimetric variables on  $Z - Z_{DR}$  space and  $Z - K_{DP}$  space. Scarchilli et al. (1996) suggested Z calibration using the self-consistency of Z,  $Z_{DR}$ , and  $K_{DP}$ . The KMA Weather Radar Center (WRC) has also suggested relations between polarimetric variables, such as the  $Z - Z_{DR}$  and  $Z - K_{DP}$  relations, using observation measurements in order to calibrate the Z and Z of its testbod radar (WRC 2014).
- to calibrate the Z and  $Z_{DR}$  of its testbed radar (WRC, 2014).

This study presents an empirical method to optimize polarimetric variables and produce more accurate dual-polarization radar rainfall estimation using radar constant altitude plan position indicator (CAPPI) data produced by different agencies. The polarimetric variables were adjusted using the WRC's relations and the accuracy of the estimated rainfall using the adjusted variables was assessed. Three radar rainfall estimation algorithms were employed to estimate rainfall using dual-


polarization radar. The method was evaluated by a comparison with raingauge observations.

This paper is composed of five sections including the introduction and conclusions. Section 2 explains the empirical method used to improve dual-polarization radar rainfall estimation. Section 3 presents the whole process of the empirical method using Yong-In Testbed (YIT) radar as well as an evaluation. Section 4 applies the empirical method to data from the Bangryung Island (BRI), Bislsan (BSL), and Sobaksan (SBS) radars, which are operated by different agencies. Section 5

#### **2** Empirical Method

concludes the paper.

The empirical method is designed to improve dual-polarization radar rainfall estimation by adjusting the observed polarimetric variables (Fig. 1). For each adjustment process, the rainfalls are estimated by three algorithms using the adjusted

polarimetric variables. The accuracy of each radar-estimated rainfall is then assessed. When all the adjustment processes are complete, the polarimetric variables that obtain the most accurate rainfall estimations are used as the optimized polarimetric variables.

The details of Steps 1 to 6 in the flowchart in Fig. 1(b) are as follows.

## Step 1:

This step derives the relations between polarimetric variables from ground measurements. The WRC installed a twodimensional video disdrometer (2DVD) at a ground observation station in Jincheon (hereafter Jincheon station). The 2DVD was installed to verify the polarimetric variables obtained by the YIT radar. The relations between polarimetric variables used in this study were derived using the 2DVD. In order to derive these relations, the WRC (2014) conducted experiments for 22 storm events that occurred during the summer of 2014. Two relations, the  $Z - Z_{DR}$  relation (Eq. (1)) and  $Z - K_{DP}$ relation (Eq. (2)), were suggested by the WRC (2014) (Fig. 2).

$$Z_{DR} = 0.153 \times Z^{0.205} \tag{1}$$

$$K_{DP} = 1.853 \times 10^{-4} \times Z^{0.781} \tag{2}$$

Here, the units of Z,  $Z_{DR}$ , and  $K_{DP}$  are mm<sup>6</sup> m<sup>-3</sup>, dB, and ° km<sup>-1</sup>, respectively.

The crosses in Fig. 2 indicate the polarimetric variables observed by the 2DVD. The two polarimetric variable relations mentioned in Fig. 1(a) are represented by black solid lines in Fig. 2. These relations are reasonable, as they are within the range of values suggested by other researchers (Vivekanandan et al., 1999; Straka et al., 2000). Thus, these relations are used as reference relations for this study.

#### Steps 2 and 3:

Step 2, which determines the observed bivariate distribution, and Step 3, which adjusts the polarimetric variables using the

reference relations, are explained together as they are closely linked. The bivariate distributions of  $Z - Z_{DR}$  and  $Z - K_{DP}$  observed by the YIT radar are shown as a hatched area in Fig. 1(a). The modes of the observed bivariate distributions are

adjusted so that they occur in the dashed region instead. It is, however, uncertain where the adjusted modes would occur on the line of the relations along the adjustment processes. If Z has no bias, the modes will vertically shift along the Y-axis. In other words,  $Z_{DR}$  and  $K_{DP}$  either increase or decrease without any adjustment to Z until the observation modes fall on the reference relation.

In other cases where Z has bias, this bias can vary because of environmental factors such as temperature or humidity that impact radar performance and measurements. Therefore, a degree of adjustment must be considered. Eleven levels of adjustment magnitude from 0 to 10 are used. For each level, Z is increased from 0 dBZ to 10 dBZ in intervals of 1 dBZ. At level 0, there is no bias in Z. In this case,  $Z_{DR}$  and  $K_{DP}$  are increased or decreased in order that the mode of the observed bivariate distribution falls on the reference relation.

# 10 Steps 4 and 5:

These steps estimate the rainfall using the adjusted variables and assess its accuracy. In order to validate the empirical method based on the WRC's relations, three radar rainfall estimation algorithms are employed in this study, as summarized in the Table 1. Algorithm R1 is considered to be the method most commonly used by hydrologists and KMA in Korea (Yoo et al., 2016). It is a conventional estimation algorithm (Marshall and Palmer, 1948). As this algorithm was derived from

- stratiform precipitation data, it has been shown to underestimate high rainfalls (Battan, 1973; Ryzhkov and Zrnić, 1996). Algorithm *R*2 was developed over 22 summer storm events by WRC (2014) as mentioned in Step 1. Algorithm *R*3 is the CSU-ICE algorithm (Cifelli et al, 2011). Algorithm *R*3 itself consists of four different algorithms, R(Z),  $R(Z, Z_{DR})$ ,  $R(K_{DP},$  $Z_{DR}$ ), and  $R(K_{DP})$  for the given ranges of the three polarimetric variables. The rainfalls estimated by these three algorithms are also estimated for each magnitude determined in Step 3.
- The accuracy of the rainfalls estimated by the three algorithms was assessed using the Eq. (3) for each magnitude. Values approaching 100 % indicate a better rainfall estimation. The normalized error (NE) quantifies the absolute error, and maximum 1 NE indicates minimized errors for both bias and random error.

$$1 - \mathrm{NE} = \left(1 - \frac{\sum |R_i - G_i|}{\sum G_i}\right) \times 100(\%) \tag{3}$$

Here,  $R_i$  is the radar-estimated rainfall (mm) for the *i*-th data pair and  $G_i$  is the raingauge rainfall (mm) for the *i*-th data pair.

## 25 Step 6:

In Step 6, the magnitude that obtains the best accuracy is selected and the polarimetric variables in this magnitude are considered to be optimized polarimetric variables.

## 3. Evaluation of the empirical method using the YIT radar

## 3.1 Input data

The YIT radar is an S-band dual-polarization radar manufactured by Enterprise Electronics Corporation (EEC). Its

5

20

specifications are summarized in Fig. 3(a). Its location is shown in Fig. 3(b), and its radar range of 240 km covers most inland South Korean territory. This section presents the overall empirical method process using the YIT radar.

In the empirical method, the primary input data for rainfall estimation was the CAPPI data of the YIT radar at 1.5 km in height. The CAPPI data was used as the main input data, because the impact of the bright band (or melting layer), which is often formed about 4–5 km in height for the cases considered in this study, can be avoided. In addition, it is assumed that

- hydrometeors at this height are purely rain because they are under the melting layer. The resolution of the CAPPI is 5 min in time and 1 km in space. Hourly accumulated rainfalls observed by 239 raingauges within 100 km of the radar range were used to assess the accuracy of the radar-estimated rainfall (Fig. 3(b)). Statistically, each raingauge covers about 131 km<sup>2</sup>, and the distance between two raingauges is approximately 11.4 km.
- 10 The three storm events used to validate the method are summarized in Table 2. Event 1 (12 July 2015) was a largely developed stratiform precipitation influenced by the typhoon *Chanhom*, which had mainly travelled over the Yellow Sea and hit mainland China, as shown on the left of Fig. 4(a). The maximum recorded hourly gauge rainfall was just 18.0 mm during Event 1 (Fig. 4(a), middle). The bright band was also clearly detected at about 5 km in height by the YIT radar, as shown on the right of Fig. 4(a).
- Event 2 (23–26 July 2016) was a frontal convective precipitation with the leading convective cells in a line from the south-west to the north-east (Fig. 4(b), left). The maximum hourly rainfall of 57.5 mm was recorded at 0200 KST 25 July 2015, as shown in the middle in Fig. 4(b). The cause was a southern cold front faced with a warm front from the north in the middle of the Korean Peninsula that stayed over 72 h, as shown in the surface weather chart on the right of Fig. 4(b).

Event 3 was a multicell superstorm accompanied by strong lightning, as shown in the left and right panels of Fig. 4(c). The convective cells developed from 1200 to 1900 KST 8 August 2015. During this time, persistent thunderstorms and contiguous precipitation areas were produced. The strongest reflectivity appeared to be 55 dBZ and was surrounded by 45–

50 dBZ. The maximum hourly rainfall was 77 mm at 1500 KST 8 August 2015.

# **3.2 Evaluation and results**

Figure 5(a) shows the observed bivariate distribution of  $Z - Z_{DR}$  during Event 3. The solid line in the figure is the  $Z - Z_{DR}$ 25 relation of Eq. (1). As shown in Fig. 5(a), a part of the distribution at high frequency (20 

# Table 3.


Figure 7 shows the radar-estimated rainfall derived with Algorithm *R*3 using the data of Table 3. As shown in the figure, the estimated rainfalls approach a one-to-one line as the polarimetric variables are adjusted, indicating that the bias has decreased. The variability is also reduced, which indicates that the random error has decreased. The best accuracy for rainfall estimation is obtained between magnitudes 6 and 8.

The accuracy of the rainfall estimations according to the eleven magnitudes is illustrated in Fig. 8. The accuracy of rainfall estimations using the observed polarimetric variables without adjustment are plotted along the Y-axis (labelled "No Adj."). Generally, the plots show increasing accuracies as the level of adjustment increases from magnitude 0 to the magnitude level that gives the best accuracy. Once each event reaches the best accuracy at a certain magnitude, then the accuracy tends to be accuracy to be adjustment for the improvement.

decrease as the level of adjustment further increases.

Event 1 almost symmetrically changes before and after the best accuracy at magnitude 5. Event 2 forms a plateau with a longer, gradually increasing upwards slope, as shown in Fig. 8(b), as there are no big changes after magnitude 7, which obtains the best accuracy. Event 3 is quite similar to Event 1, although it has a wider range of decreasing accuracy rates for magnitudes 6–10. Nevertheless, the three algorithms for all events show that a certain magnitude of adjustment produces the best rainfall astimation accuracy.

best rainfall estimation accuracy.

Table 4 summarizes the accuracy obtained by the three algorithms using the observed ("Before") and optimized ("After") polarimetric variables. The accuracy of the rainfall estimation using optimized polarimetric variables is more than 70 % accurate for most cases. For Event 2, an accuracy of a more than 75 % was obtained by all algorithms using the optimized polarimetric variables.

- For stratiform precipitation (Event 1), accuracies from 66.9 % to 71.9 % for the estimated rainfalls were obtained by all algorithms. Algorithm *R*3 does not rely much on the  $K_{DP}$  variable for Event 1; therefore, its performance is similar to the performance achieved by the algorithm using *Z* and  $Z_{DR}$  (Algorithm *R*2). Algorithm *R*3 was able to obtain a better accuracy for convective precipitation (Event 3).
- Algorithm *R*2, which was suggested by WRC, performed fairly well on all three events. The best accuracy for each event was gained using polarimetric variables adjusted using magnitudes 5 (Event 1) and 8 (Events 2 and 3). Algorithm *R*2 has a performance that is very similar to those of Algorithms *R*1 and *R*3, although each algorithm obtains the best accuracy using polarimetric variables adjusted by different magnitudes. Thus, the relations of the polarimetric variables for Algorithm *R*2 can be regarded as suitable for estimating rainfall using the YIT radar.

Hourly rainfalls estimated by Algorithm R3 using optimized polarimetric variables are compared to hourly gauge rainfalls

for the three events in Fig. 9. As shown in the figure, the radar-estimated rainfalls have nearly the best accuracy. Furthermore, their relationships with the observed rainfalls formed nearly a one-to-one line for all events. However, rainfalls underestimated by the radar still exist, as there are circle points along the X-axis. This could be caused by the attenuation of *Z* caused by partial beam blockage.

#### 4 Application of the method to radars operated by different ministries

#### 4.1 Input data

In this section, rainfall was estimated using radar data and its accuracy was assessed for the BRI radar of KMA and the BSL and SBS radars of MOLIT using the empirical method developed using the YIT radar. The  $Z - Z_{DR}$  and  $Z - K_{DP}$  relations and

- Algorithm *R*2 complemented by WRC (2015) were used (Eqs. (4)–(6)). The specifications of the three radars are summarized in Fig. 10(a) and their locations are shown in Fig. 10(b). While the observational range of the weather radar is 240 km, the observational range of the rain radar is 150 km so that the rain can be monitored at a low altitude and contamination by the bright band can be avoided. This study used automatic weather stations (AWSs) within 150 km of the radar range to assess the accuracy of the three radars under the same conditions (Fig. 10(b)). In addition, all events from May
- to October 2015 (17 events for the BRI radar, 27 events for the BSL radar, and 28 events for the SBS radar) were used for the accuracy assessment (Table 5).

$$Z_{DR} = 0.1015 + 0.00511Z + 0.00049 \times Z^2 \tag{4}$$

$$K_{DP} = 2.575 \times 10^{-13} \times Z^{7.5} \tag{5}$$

$$R2: R(Z, Z_{DR}) = 0.0081 \times Z^{0.91} \times Z_{DR}^{-4.2467}$$
(6)

### 15 4.2 Results

Figure 11 shows scatter plots for the hourly gauge rainfall and hourly rainfall estimated using the observed polarimetric variables. As shown in the figure, the rainfalls estimated by the three radars show different accuracies. The BRI and BSL radars underestimate the rainfalls using the three algorithms. Although using  $Z_{DR}$  and  $K_{DP}$  reduced the bias for the BSL radar, the correlation between the gauge rainfall and radar-estimated rainfall was lower. With regard to the SBS radar, Algorithm

*R*1 underestimated the rainfall. Algorithm *R*2 was more accurate, but the correlation between gauge rainfall and the rainfall estimated by Algorithm *R*3 was lower.

In fact, the accuracy of the rainfall estimated by Algorithm *R*2 was high, because the  $Z_{DR}$  observed by the SBS radar was generally low. Figure 12 shows the observed bivariate probability distribution of  $Z - Z_{DR}$  for the three radars for stratiform precipitation events. The bold line is the  $Z - Z_{DR}$  relation suggested by WRC (2015) (Eq. (4)). As shown in the figure, the

- $Z_{DR}$  of the BRI and BSL radars was mostly higher than the  $Z Z_{DR}$  relation (Figs. 12(a) and (b)). However, the  $Z_{DR}$  of the SBS radar was mostly lower than the  $Z Z_{DR}$  relation, even when the variables were negative (Fig. 12(c)). In Algorithm *R*2, because the exponential factor of  $Z_{DR}$  is negative, the negative  $Z_{DR}$  increased the rainfall underestimated by *Z*. Furthermore, the accuracy was also increased. However, the negative  $Z_{DR}$  of the SBS radar is an abnormal observation, as it indicates hail, which is not often observed in Korea during the summer. Another evidence of abnormal observation is the rainfall estimation
- obtained by Algorithm R3. The correlation between the gauge rainfall and rainfall estimated by Algorithm R3 was


deteriorated, because this algorithm filtered out the negative  $Z_{DR}$ . This indicates that the polarimetric variables of the rain radars also need to be adjusted.

Figure 13 shows the accuracy of the radar-estimated rainfall according to the magnitude of adjustment of the radars for the events shown in Fig. 12. As shown in the figure, for all three algorithms and radars, there is a certain magnitude of adjustment that produces the best rainfall estimation accuracy. The most accurate estimation used reflectivity alone, although it was adjusted by a greater magnitude than the dual-polarization variables.

Figure 14 shows scatter plots for hourly gauge rainfall and hourly radar-estimated rainfall using the optimized polarimetric variables. As shown in the figure, the relationships between the gauge rainfalls and radar-estimated rainfalls are better than those in Fig. 11, and a nearly one-to-one line was formed for all three algorithms and radars. The accuracies before and after

adjustment are summarized in Table 6. The accuracy before adjustment of the polarimetric variables is from 33.3 % to 57.8 %. However, the accuracy after the adjustment ranges from 65.7 % to 70.5 % and shows similar performance for all events and radar sites. These results show that the empirical method can provide more reliable rainfall estimates not only for individual radar sites but also for composite CAPPI data from multiple radar sites.

#### **5** Conclusions

In this paper, an empirical method was introduced and demonstrated for different types of precipitation at four radar sites operated by different agencies in Korea. The polarimetric variables were adjusted by the WRC's polarimetric variable relations  $(Z - Z_{DR} \text{ and } Z - K_{DP})$ . This method yields better accuracy rainfalls estimated by three algorithms using dualpolarized radar data. Further details of the results are as follows.

First, for all three algorithms for all events, a certain magnitude of adjustment produced the best rainfall estimation 20 accuracy. This means that the radars used for this study have similar ranges of polarimetric variables, even though they are made by different manufacturers and are operated and maintained using different strategies.

Second, the observed bivariate distributions between the polarimetric variables did not correspond with the reference relations. The polarimetric variables of all radars did not fit the relation, and even those of the SBS radar took on negative values, even though hydrometeors with negative  $Z_{DR}$  do not often develop or occur in Korea in the summer. Therefore, the distributions between polarimetric variables could be moved onto the reference relation line after they have been adjusted

distributions between polarimetric variables could be moved onto the reference relation line after they have using the best-performing magnitudes of adjustment for the observed polarimetric variables.

Third, the accuracy of the rainfall estimation using the optimized polarimetric variables showed about a 70 % accuracy for the YIT radar. Event 2 obtained an accuracy of more than 75 %. The accuracy of the rainfall estimated by Algorithm  $R^2$  (suggested by WRC) was similar to the accuracy of the other algorithms (Algorithms  $R^1$  and  $R^3$ ). Therefore, the relations of the polarimetric variables used in Algorithm  $R^2$  can be regarded as suitable for estimating rainfall using the YIT radar.


Fourth, the accuracy of the radars operated by different agencies ranged from 33.3 % to 57.8 % before the adjustments. The difference between the maximum and minimum accuracy was more than 20 %. However, the accuracy ranged from 65.7 % to 70.5 % after the adjustments and the difference decreased to less than 5 %. In addition, the accuracy increased to

5

approximately 70 % after the adjustment.

This study shows that the empirical method to adjust polarimetric variables using the referential relations suggested by WRC is a reliable method for overcoming measurement biases in dual-polarization radars for rainfall estimation. It will be useful for quantitatively improving the rainfall estimation of newly install radars, as establishing optimal or reliable quality control algorithms on new radars such as the YIT radar takes long time. In addition, the empirical method could be useful for improving the accuracy of radars operated by different agencies. Nevertheless, there is still much room for improvement in the method, particularly for radar measurements with partial beam blockage and severe systematic biases. Thus, this method will continue to be developed through applications to more varied precipitation types and real-time adjustment of the polarimetric variables in the near future.

## 10 Acknowledgement

This research was supported by the "Development and application of cross governmental dual-pol radar harmonization (WRC-2013-A-1)" project of the Weather Radar Center, Korea Meteorological Administration.

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
