# Peer review of "An empirical QPE method based on polarimetric variable adjustments"

_Atmospheric Measurement Techniques, 2016_

## Referee Comment (RC1) · Anonymous Referee #1 · 19 Feb 2017

Review: An empirical QPE method based on polarimetric variable adjustments

General comments: The manuscript presents a method to enhance radar based QPE through empirical relations between radar observations. Overall, this manuscript's logic is clear and well written, but considering its content, I suggest "reject" for the following reasons:

1.) The challenges in radar based QPE are mainly from the following three aspects: a.) The radar data quality control, this includes calibration, attenuation correction, partial beam blockage mitigation, non-meteorological clutter (ground clutter, sea clutter) removal and etc. 1 dBZ (0.1 dB) biased in the Z (ZDR) field could cause 10% biased rainfall rate estimation. More and more evidence shows that ZDR is over sensitive to calibration and attenuation, therefore, it should not be quantitatively used in rainfall

rate estimation. Currently, more accurate and robust rainfall rate estimation approach using specific attenuation has been develop for S-, C-, and X-band dual-polarization radars. b.) The relationship between the polarimetric radar variables and the rainfall. All the radar variables are sensitive to the drop size distribution (DSD) to some degree. Therefore, we choose different relations for stratiform, convective, and even typhoon precipitation. c.) Other related issues such as bright band correction, vertical profile of reflectivity (VPR) correction, radar coverage gaps and etc. Check the consistency between radar variables belongs to the quality control category, and should be done even before implement the radar in QPE. 2.) I did not see contributions from this work to the radar meteorology community. The key of this work is the "empirical relationships between polarimetric radar variables". This is based on the self-consistency principle, which has been discussed in tons of journal publications. Even the relations in Equations 1 and 2 are from WRC (2014). I think this manuscript is OK to be used as a work report, but not for a journal publication. 3.) I believe steps 2 and 3 together with figure 1 are the core part of this work, but to be honest, I have no idea what authors did after reading this paragraph and figure. Everything looks very vague. I even do not understand the Fig. 1: why the y axis is "Zdr or Kdp"? what is the value of x axis (Z), and y axis ("Zdr or Kdp")? What is the line, what is the dashed line contour? What is the star? ….

Overall, this work reports a well-accepted approach in radar based QPE. It does reveal any new findings and relationships. Its contribution is trivia. Therefore, I don't think it is ready to be published in the current shape.

---

## Referee Comment (RC2) · Anonymous Referee #2 · 6 May 2017

This paper proposes an empirical calibration of radar polarimetric variables for quantitative precipitation estimation (QPE) in Korea. Relation between polarimetric moments are obtained from disdrometer measurements. Several QPE retrievals are applied from the radar observations. The benefit of using dual polarized X-band radar data is proved useful compared to standard reflectivity-to-rainfall based algorithm estimator.

The research area of using radar data to obtain good quality distributed rain estimations recovers a real need for hydrological applications (flash-floods monitoring) and is still opened. Although the paper is structured and the objectives appear clearly, the methodology, results and operational implication should be better described.

1- Several challenges affect radar QPE besides calibration of radar moments: ground clutter, beam blockage in complex terrain, vertical structure of precipitation, partial

beam filling... A discussion of these issues and how they are handled in the present study is required.

2- The methodology is not clear on how the adjustment of radar moments is performed and what constraints are used.

3- Are the ground measurements from one disdrometer used to derive relations between polarimetric variables representative spatially and temporally (only one summer used)? Could we expect any variations in disdrometer-derived relations according to the precipitation microphysics associated with different seasons or rain types (convective, stratiform) for example?

4- How are these disdrometer-derived relations affected by the large resolution difference between the disdrometer and the radar observations?

5- Comparison of radar estimates with raingauge measurements provides indication of the pertinence of the different processing techniques employed. However the evaluation is not well developed, for example classical criteria are not used (e.g. bias, correlation, root mean square error). The abstract mentions an improvement but only a number is provided without details on the score used to assess this improvement.

6- There is no mention of the sampling difference between radar and raingauges. For comparison it would be useful to give some elements about the spatial representativity of raingauges measurements at the considered time step, and how they match the QPE spatial resolution.

- 7- Regarding the raingauge data, is data control performed?
- 8- A discussion on the operational applicability of this method would be welcome.

---

## Author Comment (AC1) · 2 Jun 2017

**Overall Response:** The reviewer commented 3 general comments. We sincerely thank you for the comments that help to improve our paper. The responses to the comments are as follow.

**Comment 1a):** *The challenges in radar based QPE are mainly from the following three aspects: a.) The radar data quality control, this includes calibration, attenuation correction, partial beam blockage mitigation, non-meteorological clutter (ground clutter, sea clutter) removal and etc. 1 dBZ (0.1 dB) biased in the Z (ZDR) field could cause 10% biased rainfall rate estimation. More and more evidence shows that ZDR is over sensitive to calibration and attenuation, therefore, it should not be quantitatively used in rainfall rate estimation. Currently, more accurate and robust rainfall rate estimation approach using specific attenuation has been develop for S-, C-, and X-band dual-polarization radars.*

**Response 1a):** QPE can be affected by the radar data quality control including calibration, attenuation correction, partial beam blockage mitigation, non-meteorological clutter removal and etc. as your comment. The input data in this study therefore was made using post quality control (QC) processed data which is removed ground clutter and identification of non/meteorological echoes by WRC's QC method based on fuzzy algorithms. So we will give a description about the sources error and the QC method in second paragraph of section 3.1 like below. However, the method in this study cannot solve the beam blockage and we give a description about the beam blockage in last paragraph of section 3.2 (page 6). We also agree that ZDR is so sensitive that it is not easy to handle the variable. The conventional variable, Z, is more stable variable for estimating the radar rainfall. However, it needs Z as well as other variables to classify hydrometeors and better estimate the radar rainfall. Also, it needs to adjust the variables together because the variables from raw data are out of proper domain in Z-ZDR or Z-KDP spaces. The empirical method in this study is designed to improve dual-polarization radar rainfall estimation by adjusting the variables

**Revision 1a):** (Second paragraph of section 3.1) QPE can be affected by ground clutter, beam blockage, vertical structure of precipitation in the case of stratification, beam filling, etc. The input data in this study therefore was made using post quality control (QC) processed data which is removed ground clutter, corrected beam blockage and identification of non/meteorological echoes by WRC's QC method based on fuzzy algorithms (WRC, 2015).

(First paragraph of section 4.1) Like the YIT radar data, the BRI, BSL and SBS radar data was also data quality controlled by WRC's QC method based on fuzzy algorithms, which includes removal of ground clutter, correction of beam blockage and identification of non/meteorological echoes (WRC, 2015).

**Comment 1b):** *b.) The relationship between the polarimetric radar variables and the rainfall. All the radar variables are sensitive to the drop size distribution (DSD) to some degree. Therefore, we choose different relations for stratiform, convective, and even typhoon precipitation.*

**Response 1b)** Recently, precipitation pattern in Korea changes due to climate change. Therefore KMA installed the first 2DVD in 2014 to observe the change of precipitation microphysics and obtain the polarimetric variable relation. KMA will continuously develop the polarimetric relation in the mid-latitude region. The relations in section 3 were derived from the 2DVD data during only one summer (22 storms) and the relations in section 4 were complemented by adding the 2DVD data (73 storms). Naturally, the variability due to the rain type can occur because the relations were derived from only one or two year data and only one point data. More relations according to the rain type have to be derived by installing more 2DVD and accumulating the 2DVD data to solve this problem. Also, it needs to examine the variability due to the rain type in the future because the 2DVD in Korea is installed recently and the data is also not enough. We agree that the variability due to the rain type can occur as your comment. So, we will add below sentence in conclusions.

**Revision 1b):** The variability due to the rain type can occur because the relations were derived from only one or two year data and only one point data. More relations according to the rain type have to be derived by installing more 2DVD and accumulating the 2DVD data to solve this problem. Also, it needs to examine the variability due to the rain type in the future because the 2DVD in Korea is installed recently and the data is also not enough.

**Comment 1c):** *c.) Other related issues such as bright band correction, vertical profile of reflectivity (VPR) correction, radar coverage gaps and etc. Check the consistency between radar variables belongs to the quality control category, and should be done even before implement the radar in QPE.*

**Response 1c):** We gave a description about bright band in second paragraph of section 3.1 like below.

**Revision 1c):** (Second paragraph of section 3.1) In the empirical method, the primary input data for rainfall estimation was the CAPPI data of the YIT radar at 1.5 km in height. The CAPPI data was used as the main input data, because the impact of the bright band (or melting layer), which is often formed about 4–5 km in height for the cases considered in this study, can be avoided. In addition, it is assumed that hydrometeors at this height are purely rain because they are under the melting layer.

**Comment 2):** *I did not see contributions from this work to the radar meteorology community. The key of this work is the "empirical relationships between polarimetric radar variables". This is based on the self-consistency principle, which has been discussed in tons of journal publications. Even the relations in Equations 1 and 2 are from WRC (2014). I think this manuscript is OK to be used as a work report, but not for a journal publication.*

**Response 2):** We agree that this study is based on the self-consistency principle. The self-consistency principle (ex. Scarchilli et al., 1996) calculates the reflectivity error from comparing observed differential phase shift with the estimated differential phase shift based on the reflectivity and differential reflectivity measurements. If the observed differential phase shift is really perfect, the observed differential phase shift could be truth. In field work, however, we doubt that the differential phase shift is perfect. Therefore this study adjusts the radar variables and finds the optimized variables using the gauge on the ground. We thought that this method can be very useful because the empirical method can estimate the radar rainfall quantitatively and qualitatively similar to the gauge rainfall. This study is based on empirical and technical method. Therefore we submit our paper in this journal.

**Revision 2):** There is no revision.

**Comment 3)** *Dataset. 3.) I believe steps 2 and 3 together with figure 1 are the core part of this work, but to be honest, I have no idea what authors did after reading this paragraph and figure. Everything looks very vague. I even do not understand the Fig. 1: why the y axis is "Zdr or Kdp"? what is the value of x axis (Z), and y axis ("Zdr or Kdp")? What is the line, what is the dashed line contour? What is the star?*

**Response 3)** We have been working on enhancing the section 2 with further detail explanation and figures. Also the constrain of the method is explained in the section.

**Revision 3)**

(1) We will revise the figure 1 and also add below sentence.

 → Fig. 1 show each step of adjustment process in the empirical method.

[Figure]

**Figure 1**. Flow chart of empirical method

(2) We changed a word, 'derives' to 'selects' in Step 1. In fact, we just select derived reference lines in Step 1. Any reference lines can be selected but we choose the reference lines to reflect microphysics of precipitation in Korea. So we used the reference lines derived from the 2DVD in Korea. Therefore we will revise the first paragraph in Step 1 of section 2.

 → This step selects the relations between polarimetric variables from ground measurements. The WRC installed a two-dimensional video disdrometer (2DVD) at a ground observation station in Jincheon (hereafter Jincheon station). The 2DVD was installed to verify the polarimetric variables obtained by the YIT radar as well as to define microphysics of precipitation in Korea, particularly the its change due

to climate change which has already shows changes on occurrence, intensity and features of precipitation, specifically during summer. The relations between polarimetric variables used in this study were derived using the 2DVD based on the first year observation. In order to derive these relations, the WRC (2014) conducted experiments for 22 storm events that occurred during the summer of 2014. Two relations, the $Z - Z_{DR}$ relation (Eq. (1)) and $Z - K_{DP}$ relation (Eq. (2)), were suggested by the WRC (2014) (Fig. 2). Any relations can be selected but below relations that reflect microphysics of precipitation in Korea are selected in this study.

(3) We will revise Steps 2 and 3 of section 2 and add a figure as Fig. 3.

-> Step 2, which determines the observed bivariate distribution, and Step 3, which adjusts the polarimetric variables using the reference relations, are explained together as they are closely linked. Fig. 3 is a schematic diagram which show how to adjust the polarimetric variables. First, two bivariate distributions of $Z - Z_{DR}$ and $Z - K_{DP}$ observed by the radar are determined as a hatched area in Fig. 3(a). Next, the most frequent value (mode) in the observed bivariate distribution which is the mark of X in the hatched area has to be found. Then, the bivariate distributions move but the modes are constrained to be on the reference relations so that they occur in the dashed region.

It is, however, uncertain where the adjusted modes would occur on the line of the relations along the adjustment processes. Therefore, a degree of adjustment must be considered. Eleven levels of adjustment magnitude from 0 (M0) to 10 (M10) are used. At level M0, there is no bias in $Z$ and the modes will vertically shift along the Y-axis (Fig. 3(b)). In this case, $Z_{DR}$ and $K_{DP}$ are either increased or decreased in order that the mode of the observed bivariate distribution falls on the reference relation. In other cases where $Z$ has bias, this bias can vary because of environmental factors such as temperature or humidity that impact radar performance and measurements. In this case, $Z$ is increased from 1 dBZ (M1) to 10 dBZ (M10) in intervals of 1 dBZ and also $Z_{DR}$ and $K_{DP}$ are either increased or decreased (Fig. 3(c)).

[Figure]

**Figure 3**. Schematic diagram of observed bivariate distribution (left panel) and bivariate distribution shift (middle panel: Z has no bias, right panel: Z has bias): (a) $Z - Z_{DR}$ space and (b) $Z - K_{DP}$ space

---

## Author Comment (AC2) · 2 Jun 2017

**Overall Response:** We sincerely thank you for the comments that help to improve our paper. The responses to the comments are as follow.

- **Comment 1):** Several challenges affect radar QPE besides calibration of radar moments: ground clutter, beam blockage in complex terrain, vertical structure of precipitation, partial beam filling. . . A discussion of these issues and how they are handled in the present study is required.
- Response 1) In this study, the ground clutter was removed by WRC's QC method based on fuzzy algorithm. We also cannot avoid the problem of the beam blockage because 70% of Korean territory is covered with mountains. However, the method in this study cannot solve the beam blockage and we give a description about the beam blockage in line 18 ~ 20 of page 7 and line 1 of page 10. Of problems caused by vertical structure of precipitation, the bright band of stratiform rain can overestimate radar rainfall. In this study, we try to avoid this problem by using 1.5km CAPPI and describe why we use 1.5km CAPPI in second paragraph of page 5. The problem of partial beam filling can be handled in the signal process. We cannot handle the partial beam filling because this study used the moment data after the signal process. We agree your comment that above sources of error can affect radar QPE. So we will give a description about the sources error and the QC method like below.
- **Revision 1a**): (Second paragraph of section 3.1) QPE can be affected by ground clutter, beam blockage, vertical structure of precipitation in the case of stratification, beam filling, etc. The input data in this study therefore was made using post quality control (QC) processed data which is removed ground clutter, corrected beam blockage and identification of non/meteorological echoes by WRC's QC method based on fuzzy algorithms (WRC, 2015). In the empirical method, the primary input data for rainfall estimation was the CAPPI data of the YIT radar at 1.5 km in height. The CAPPI data was used as the main input data, because the impact of the bright band (or melting layer), which is often formed about 4–5 km in height for the cases considered in this study, can be avoided. In addition, it is assumed that hydrometeors at this height are purely rain because they are under the melting layer.
  - (First paragraph of section 4.1) Like the YIT radar data, the BRI, BSL and SBS radar data was also data quality controlled by WRC's QC method based on fuzzy algorithms, which includes removal of ground clutter, correction of beam blockage and identification of non/meteorological echoes (WRC, 2015).

- **Comment 2):** The methodology is not clear on how the adjustment of radar moments is performed and what constraints are used.
- **Response 2):** We have been working on enhancing the section 2 with further detail explanation and figures. Also the constraint of the method is explained in the section.

**Revision 2):**

(1) We will revise the figure 1 and also add below sentence.

-> Fig. 1 show each step of adjustment process in the empirical method.

| Step 1: Selection of relations between polarimetric variables                       |  |  |  |  |  |  |
|-------------------------------------------------------------------------------------|--|--|--|--|--|--|
| └                                                                                   |  |  |  |  |  |  |
| Step 2: Determination of observed bivariate distribution                            |  |  |  |  |  |  |
| ↓                                                                                   |  |  |  |  |  |  |
| Step 3: Adjustment of polarimetric variables using relations with the eleven levels |  |  |  |  |  |  |
| ▼                                                                                   |  |  |  |  |  |  |
| Step 4: Estimation of radar rainfall using adjusted variables each level            |  |  |  |  |  |  |
| ▼                                                                                   |  |  |  |  |  |  |
| Step 5: Assessment of accuracy of radar rainfall estimation each level              |  |  |  |  |  |  |
| ↓                                                                                   |  |  |  |  |  |  |
| Step 6: Selection of optimized variables with best accuracy                         |  |  |  |  |  |  |

Figure 1. Flow chart of empirical method

- (2) We changed a word, 'derives' to 'selects' in Step 1. In fact, we just select derived reference lines in Step 1. Any reference lines can be selected but we choose the reference lines to reflect microphysics of precipitation in Korea. So we used the reference lines derived from the 2DVD in Korea. Therefore we will revise the first paragraph in Step 1 of section 2.
  - -> This step selects the relations between polarimetric variables from ground measurements. The WRC installed a two-dimensional video disdrometer (2DVD) at a ground observation station in Jincheon (hereafter Jincheon station). The 2DVD was installed to verify the polarimetric variables obtained by the YIT radar as well as to define microphysics of precipitation in Korea, particularly the its change due to climate change which has already shows changes on occurrence, intensity and features of precipitation, specifically during summer. The relations between polarimetric variables used in this study were derived using the 2DVD based on the

first year observation. In order to derive these relations, the WRC (2014) conducted experiments for 22 storm events that occurred during the summer of 2014. Two relations, the  $Z - Z_{DR}$  relation (Eq. (1)) and  $Z - K_{DP}$  relation (Eq. (2)), were suggested by the WRC (2014) (Fig. 2). Any relations can be selected but below relations that reflect microphysics of precipitation in Korea are selected in this study.

- (3) We will revise Steps 2 and 3 of section 2 and add a figure as Fig. 3.
  - -> Step 2, which determines the observed bivariate distribution, and Step 3, which adjusts the polarimetric variables using the reference relations, are explained together as they are closely linked. Fig. 3 is a schematic diagram which show how to adjust the polarimetric variables. First, two bivariate distributions of  $Z - Z_{DR}$  and  $Z - K_{DP}$  observed by the radar are determined as a hatched area in Fig. 3(a). Next, the most frequent value (mode) in the observed bivariate distribution which is the mark of X in the hatched area has to be found. Then, the bivariate distributions move but the modes are constrained to be on the reference relations so that they occur in the dashed region.

It is, however, uncertain where the adjusted modes would occur on the line of the relations along the adjustment processes. Therefore, a degree of adjustment must be considered. Eleven levels of adjustment magnitude from 0 (M0) to 10 (M10) are used. At level M0, there is no bias in Z and the modes will vertically shift along the Y-axis (Fig. 3(b)). In this case,  $Z_{DR}$  and  $K_{DP}$  are either increased or decreased in order that the mode of the observed bivariate distribution falls on the reference relation. In other cases where Z has bias, this bias can vary because of environmental factors such as temperature or humidity that impact radar performance and measurements. In this case, Z is increased from 1 dBZ (M1) to 10 dBZ (M10) in intervals of 1 dBZ and also  $Z_{DR}$  and  $K_{DP}$  are either increased or decreased or decreased or decreased (Fig. 3(c)).